# Experimental Study of Pyrolysis and Laser Ignition of Low-Vulnerability Propellants Based on RDX

**DOI:** 10.3390/molecules25102276

**Published:** 2020-05-12

**Authors:** Jordan Ehrhardt, Léo Courty, Philippe Gillard, Barbara Baschung

**Affiliations:** 1PRISME EA 4229, Université d’Orléans, 18020 Bourges, France; Jordan.EHRHARDT@isl.eu (J.E.); philippe.gillard@univ-orleans.fr (P.G.); 2French-German Research Institute of Saint-Louis, 68300 Saint-Louis, France; Barbara.BASCHUNG@isl.eu

**Keywords:** pyrolysis, ignition energy, combustion characteristics, RDX, nitrocellulose, low-vulnerability propellant

## Abstract

Low-vulnerability propellants are propellants designed to resist unintended stimuli to increase safety during transport, storage and handling. The substitution of usual nitrocellulose-based gun propellants with these new materials allows maintaining interior ballistics performances while increasing the safety. In this paper, the pyrolysis, ignition and combustion of such propellants are investigated in order to study conditions leading to a safe and reproducible ignition. Low-vulnerability propellants studied are made of different ratios of hexogen (RDX) and nitrocellulose (NC). Three compositions are studied by varying weight percentages of RDX and NC: 95-5, 90-10 and 85-15 for respective weight percentages of RDX-NC. Pyrolysis of these propellants is studied with two different experimental setups: a flash pyrolysis device linked to a gas chromatograph coupled to a mass spectrometer (Py-GC-MS) and a closed-volume reactor coupled to a mass spectrometer. Different molecules, like NO_2_, CO, CH_3_COCH_3_ or CH_2_NCH_2_NCH_2_, are obtained during the decomposition of these propellants. Laser ignition of these propellants is studied in a cylindrical closed-volume reactor using a laser diode. Several combustion characteristics, such as ignition delays, maximal overpressures and combustion rates are given for the three propellants using the pressure signals. Moreover, ignition energies are also investigated. Obtained results are compared to the few available literature data. A particular behavior is noticed for the 90-10 propellant. The experimental data collected should serve in the future to have a better understanding of the chemical reactions driving the combustion process of these low-vulnerability propellants.

## 1. Introduction

Propellants are energetic materials known for their use in space propulsion fields and more widely when high pressures and energies are required with low operating times. One of the main prerequisites for field application is the performance of the propellant. In recent years, two other factors have been included in the elaboration of energetic materials. On the one hand, the impact of the propellant on the environment should be investigated. Thus, environmentally friendly propellants have been developed. They also have to comply with REACh regulations. On the other hand, safety during handling, transportation and storage should be increased. This aspect led to Low Vulnerability Ammunition (LOVA) propellant. Indeed, the use of energetic materials induced accidental situations including materials destructions or damage to human lives. This work aims at investigating pyrolysis and laser ignition of such low vulnerability propellants.

During the last sixty years, several researches have been conducted to study and model propellants combustion and their laser ignition. Pioneering work on laser ignition of energetic materials was performed by Price et al. [1] in 1966. They published a review of the main research achieved dealing with propellants ignition. Later in 1982, some reviews of experimental and numerical works have been published by both Kulkarni et al. [2] and Hermance [3]. In 1984, a complete review of combustion propellants properties was written by Boggs [4]. Twenty-two years later, Beckstead [5] reviewed the different kinetic models existing and detailed the numerical schemes used. Propellant characteristics, such as combustion rates, are presented in the work of Atwood [6]. The same authors also estimated the sensitivity of the combustion rate toward temperature [7]. More recently, De et al. [8] investigated high nitrogen materials ignition with a CO_2_ laser. Their work focused on the ignition delays and energies. Ignition of elastomer-modified cast double-base propellant has been published by Herreros and Fang in 2017 [9]. Their investigations detailed ignition delays, burning times and compared granular to strand propellants. Commercial low-vulnerability propellants laser ignition has also been studied by Gillard et al. [10] and Courty et al. [11]. They published some numerical and experimental results concerning a hexogen (RDX) based and a nitrocellulose (NC) based propellant.

Pyrolysis of propellants has also been studied over the past decades, but there are few works on nitrocellulose available in the literature. In 1987, Beck [12] investigated the pyrolysis of different polymeric binders used for propellants. Some investigations on nitrocellulose and RDX pyrolysis were also published by Huewei et al. [13,14]. They detailed the major chemical species found through their flash pyrolysis apparatus coupled with a gas chromatography and mass spectrometer. Concerning the RDX pyrolysis, they concluded that the major nitrogen oxide in the RDX pyrolysates is not NO_2_ but NO. These data can provide important information for numerical modelling or to build a decomposition mechanism of such energetic materials.

A review on design and technologies for insensitive munitions was performed by Powell [15]. Physical characteristics of gun propellants, along with their compositions, can play a role in the sensitivity to accidental or voluntary stimuli. More precisely, the reduction of the particle sizes significantly reduces the shock sensitivity. Concerning the composition, reduced sensitivity RDX appears to be a good candidate to reach LOVA requirements. We can also note the very recent study of Jaiswal et al. [16] who formulate an enhanced energy propellant based on RDX and NC with better stability and thermal properties than conventional propellants.

In the present study, the pyrolysis, ignition and combustion of three low-vulnerability propellants made of nitrocellulose and RDX are investigated with three experimental setups. On the one hand, propellant’s pyrolysis is studied using two techniques: by means of a laser diode with an in-situ analysis using a mass spectrometer, and with a flash pyrolysis device linked to a gas chromatograph coupled to a mass spectrometer (Py-GC-MS). On the other hand, ignition and combustion are investigated in a closed-volume reactor. The originality of this work is that it presents both emitted gaseous species and combustion characteristics of three propellants with different RDX-NC ratios.

Section 2 presents results and discussion of the laser induced pyrolysis, of the Py-GC-MS experiments and of the laser ignition experiments. Section 3 describes the investigated propellants and the experimental setups. Finally, Section 4 concludes this paper.

## 2. Results and Discussion

### 2.1. Laser-Induced Pyrolysis

Preliminary experiments were performed in order to screen the mass of the species to be followed. Details on these preliminary experiments can be found in [17]. Figure 1 summarizes the atomic mass units followed as functions of time with the MS and their corresponding species, identified using the studies of Huewei et al. [13,14].

Table 1 presents the relative intensities for the selected atomic mass units regarding the gas nature under 1 bar and the composition of the propellant. Signals have been normalized regarding the intensity of pressurization gas (28 for N_2_ and 40 for Ar). As some species are detected with the secondary electron multiplier (SEM), they can present intensities higher than the one of the surrounding gas, leading to relative intensities higher than one. Despite this trend, it is possible to compare species detected with SEM detector between them.

Raw intensities show thatpyrolysis under argon at a given initial pressure provides the highest intensities of measurements. It can be noticed that for *m/z* = 28, the highest intensities are measured under nitrogen atmosphere. Indeed, N_2_ has a molecular mass of 28 g·mol^−1^ which enhances the intensity at this amu.

The highest peak intensities (about 5.3 times) are recorded under argon atmosphere. This enhancement can be explained by the differences in molar heat capacities of these gases. In fact, argon has a higher heat capacity at constant volume (Cv) than nitrogen for all temperatures. For instance, the ratio Cv(N_2_)/Cv(Ar) is of 1.66 at 0 °C and of 1.82 at 600 °C.

Measured data show that 90-10 propellant pyrolysis provides more products than the other ones for amu below 46 (CO, NO, HCHO, CO_2_, N_2_O, CH_3_CHO, NO_2_). Above this amu, the 85-15 composition emits more pyrolysis gases. It can be noticed that obtained species are similar to the ones given by Huwei et al. [13,14].

The influence of initial pressure on the emitted species is presented in Figure 2, with the example of the 90-10 propellant under nitrogen atmosphere. To study this influence, it is necessary to divide each peak height by the highest peak height, to obtain a relative intensity *Ir*. Indeed, when pressure is increased speed is also increased, a bigger amount will be analyzed and peak intensities cannot be confronted.

We can notice in Figure 2 that all values are higher than one, meaning that relative species concentrations emitted under 3 bar of nitrogen are higher than the ones emitted under 1 bar. Initial pressure is therefore a major factor in the pyrolysis process. Similar results are obtained under argon. Let us note that *m/z* = 28 presents an increased intensity between 1 and 3 bar (quotient equal to one in Figure 2 since it shows the highest peak height under nitrogen).

It is also interesting to notice the increase in the HCN peak (amu = 27) for the experiment performed under 3 bar of nitrogen. This peak intensity is not recorded during the experiment performed under argon. Under an initial pressure of 3 bar, nitrogen enhances the formation of hydrogen cyanide while argon does not.

### 2.2. Flash-Pyrolysis

With the Py-GC-MS parameters presented in the Materials and Methods sections, four peaks are observed for retention times of 7.32, 7.92, 8.77 and 11.71 min, respectively. Peaks on the GC chromatograph can be correlated to their chemical species by means of the mass spectrometer using the NIST library, as shown in Figure 3. The lightest compounds such as CO, NO, N_2_O and NO_2_ are the first to be analyzed through a flame ionization detector (FID) (peak 1). Afterwards, other chemical species such as HCN along with CH_2_O (peak 2) and then C_2_H_2_O_4_ (oxalic acid, peak 3) are separated through the columns. The last molecule to be enlightened by the mass spectrometer is C_3_H_3_N_3_ (peak 4) known as the 1,3,5-triazine resulting from the decomposition of RDX.

It can be noticed in Figure 4 that the probe temperature has a huge influence on the pyrolysis products. Indeed, the area for the 7.32 min peak is getting bigger when the probe temperature rises (about 1.65 times). Concerning the species found for a retention time of 11.7 min, the opposite trend can be observed. For a 300 °C temperature probe, high molecular mass species are the major products of the pyrolysis, whereas for higher temperatures small amu species are the major products.

GC chromatographs of the three propellants are compared in Figure 5 for a probe temperature of 500 °C. The highest peak intensity is obtained for the 90-10 composition concerning the light compounds. Regarding the high molecular mass species, the 85-15 propellant is the one giving the highest intensity.

High temperature enhances the decomposition of the RDX molecule to obtain lighter molecules such as HCN, NO_2_ or NO. For each probe temperature, the 90-10 propellant is the one providing the biggest area for the 3 first peaks (7.32, 7.92, 8.77 min) and the smallest for the last one. An optimal rate between RDX and NC may be found due to this unexpected behavior of the 90-10 composition. Indeed, lighter products are found, which may lead to an easier ignition and better combustion properties.

### 2.3. Laser Ignition

Results are detailed in this paper for argon atmosphere, detailed results under nitrogen can be found in [17].

Overpressures as functions of initial pressure under argon atmosphere are presented in Figure 6 for the three studied propellants. The uncertainty for the initial pressure is estimated at 2.5 bar, whereas the standard deviation for ignition delay is calculated from three different experiments. As the mass of the samples can vary, the overpressure was divided by the mass of the propellant. First, it can be noticed that the overpressure increases with the initial pressure. The same trend has already been reported in the literature [10,11]. Regarding the influence of the NC rate, some interesting facts can be observed. The formulation giving the highest overpressures, for a given initial pressure, is the 90-10 composition. The data presented in Figure 6 are fitted with a linear law: ∆P = a × P0 + b.

The propagation rate is estimated by calculating $\frac{1}{m}.{\left(\frac{d\left(\Delta P\right)}{dt}\right)}_{max}$.

The ignition delay can be defined as the time between the beginning of the laser beam and the beginning of the pressure increase. Figure 7 and Figure 8, respectively, present ignition delay and propagation rates as functions of initial pressures for argon atmosphere.

It is clear in Figure 7 that ignition delay decreases when initial pressure increases. The influence of the NC rate is clearly visible. As already noticed with overpressures, 90-10 propellant has a unique behavior. Indeed, it shows the lowest ignition delays at low pressures and the highest at high pressures. This irregular behavior might be the results of the white surface of the propellant that could reflect the laser beam and influence ignition delays results. Nevertheless, the formulation giving the highest ignition delay is the 95-5 and the lowest is obtained for the 85-15. The trend of these results was expected.

Propagation rates presented in Figure 8 show the same trend as overpressures. They strongly increase between 30 and 40 bar, whereas the increase is slighter above 50 bar. The 90-10 composition shows the highest propagation rate, as previously mentioned for overpressures. It can be assumed from these trends that an optimum rate between RDX and NC might exist to enhance propellant combustion properties. Interestingly, it was also noticed with the Py-GC-MS results.

The influence of the atmosphere is presented in Figure 9, Figure 10, Figure 11 and Figure 12 on the propellant 90/10. Putting together several pellets in the poly(methyl methacrylate (PMMA) holder, it is possible to measure overpressures for several initial masses of propellant. This is presented in Figure 9 for an initial pressure of 50 bar. It is clear that overpressure has a linear evolution as a function of initial mass. Obtained equations are 0.169*m* + 1.0831 (R^2^ = 0.995) for Ar and 0.108*m* + 1.777 (R^2^ = 0.998) for N_2_, *m* being the mass in mg. It is also clear that obtained overpressures are higher under Ar than under N_2_ atmosphere. This result has already been noted in the literature for commercial low-vulnerability propellants [10,11].

Overpressures, propagation rates and ignition delays are presented in Figure 10, Figure 11 and Figure 12, respectively, as functions of initial pressures for the 90-10 propellant and the two studied atmospheres. It can be noticed that overpressures and propagation rates are increasing with the initial pressure for the two studied atmospheres. Values of combustion properties under argon are higher than the ones under nitrogen for each initial pressure. For instance, propagation rates under argon are on average 1.25 time higher than under nitrogen. The effect of surrounding gas on these parameters can be enlightened by the molar heat capacities differences between nitrogen and argon. The heat capacity (Cv) of argon is indeed lower than that of nitrogen for all temperatures.

No significant differences between ignition delays under nitrogen or argon can be noticed. The same trends are reported for RDX and NC based propellant [10,11]. We can notice that for the lowest pressure, delays are longer under nitrogen, and the contrary for the other pressures. Values are fitted with an exponential law t_i_ = a × exp(b × P_0_), coefficients values are given in Table 2.

The influence of laser power is presented in Figure 13 and Figure 14. Figure 13 presents the ignition delay as a function of the laser power for the three studied propellants, at an initial pressure of 50 bar of nitrogen. The influence of the laser power is clearly visible in this figure. As expected, it can be seen that the ignition delay increases when the laser power decreases. Results show the same trend concerning the nitrocellulose rate. For each laser power studied, the ignition delay increases when the rate of nitrocellulose decreases. Figures are not presented here, but the influence of laser power on overpressure or propagation rate is very low [18,19]. Indeed, these two parameters are combustion characteristics, not dependent on ignition condition. On the contrary, ignition delay is an ignition characteristic, therefore depending on ignition conditions.

Figure 14 presents the energy giving an ignition probability of 50% (E_50_) as a function of laser power for the three studied propellants under argon atmosphere. Such energies are measured using the Langlie method [20], and more precisely with the “revised” method given by the GTPS [21]. This approach is based on the dichotomic principle and assumes a statistical repartition of the ignition threshold through normal distribution. Using this approach, the 50% ignition probability can be obtained with around 26 shots.

The first noticeable fact reading Figure 14 is the high value of E_50_ compared to the ones for RDX based propellants determined by Courty et al. [11]. These differences can be explained by the fact that the propellants studied here are not commercial propellants. Propellants samples are white as they only contain two energetic materials and a stabilizer. In commercial propellants, several additives such as graphite can be added to optimize the propellants characteristics.

The effect of the laser power on the E_50_ is clearly visible in Figure 14. The higher the laser power, the lower E_50_ is. For each propellant, the standard deviation decreases when the laser power increases. This has already been noted in the literature for propellant laser ignition [10,22]. This can be explained because increasing the power leads to an increase in the heated volume. The bigger the hot spot is, the better the homogeneity is, which leads to a decrease in result discrepancy.

The 85-15 propellant formulation presents the lowest E_50_ for the highest powers studied (10.13 and 8.74 W). For high powers, the 90-10 propellant might present higher E_50_ values than 95-5, but the standard deviations for these two propellants have the same order of magnitude. Consequently, it appears difficult to compare them, 90-10 E_50_ values might be just higher or lower than 95-5 ones.

These results show the important effect of the NC in an energetic material composition. Indeed, high nitrocellulose rates lead to high sensitivities. Here again, we can notice the unexpected behavior of the 90-10 propellant, which presents the lowest E_50_ for low powers. Similar comments are possible here concerning the standard deviations as they seem similar for 95-5 and 90-10 propellants.

## 3. Materials and Methods

### 3.1. Materials

Propellants studied in this paper are laboratory propellants produced at the French-German Research Institute of Saint-Louis (ISL, Saint-Louis, France), their composition is perfectly controlled. They consist of a tubular pellet made of nitrocellulose (NC), hexogen (RDX) and Centralit I (1,3-Diethyl-1,3-diphenylurea) as stabilizer. Nitrocellulose used in this study is 11.7% nitrated. In order to investigate the nitrocellulose rate in the propellant, three formulations were prepared:

RDX (94.7%)–5% NC–0.3% Centralit I, named as 95-5;

RDX (89.7%)–10% NC–0.3% Centralit I, named as 90-10;

RDX (84.7%)–15% NC–0.3% Centralit I, named as 85-15.

The average mass, length and diameter of the tubular pellets were, respectively, 40 mg, 5 mm and 3 mm. The inside hole diameter was of the order of 0.5 mm, as shown in Figure 15.

### 3.2. Laser-Induced Pyrolysis Setup

This setup consists of a stainless-steel closed reactor adapted to collect and analyze the pyrolysis gases. This experimental setup is most commonly used to investigate laser ignition of propellants [10,11]. A mass spectrometer (MS) Pfeiffer Vacuum—GSD 301C (Pfeiffer Vacuum, Asslar, Germany), is connected into the analysis chamber at the back of the reactor. Pellets pyrolysis is obtained with a Coherent laser diode (FAP-I-P1396, (Coherent, Palo Alto, CA, USA)) operating at 808 nm. By means of an optical system consisting of two lenses of 16 and 25 mm focal lengths, the laser spot on the propellant sample has a diameter of 1.25 mm.

The heating rate of the sample surface is estimated through an analytic modelling. This modelling relies on the assumption of a 1D thermal diffusion inside an inert material with infinite depth and laser absorption only at the surface. The analytic solution is well known and gives temperature evolution in depth and for each time from the beginning of the heating. At the surface we have:(1)$$T_{RDX-NC}=\frac{2P_{0}}{\lambda_{RDX-NC}}\times \sqrt{\frac{a_{RDX-NC}t}{\pi }}$$
where *Po* is the ratio of the absorbed laser power on the surface spot, *λ* the thermal conductivity and *a* the thermal diffusivity. These last two parameters are obtained through a mixing law of pure constituents with ponderation with mass fraction. With the incoming power of 10.13 W and the dimension of the laser spot (1.25 mm), we thus obtained 8000 °C·s^−1^ for the heating rate. Different initial atmospheres and pressures were tested in this study: nitrogen and argon under two initial pressures of 1 and 3 bar. Higher initial pressures have not been tested because of the mass spectrometer analysis chamber pressure (5 × 10^−6^ mbar). Higher pressure would have led to increase the analysis chamber pressure which could have caused the deterioration of the MS filament.

Experiments are conducted as follows: a propellant sample is inserted in a poly(methyl methacrylate) (PMMA) holder placed in the reactor and sealed with a sapphire window. Initial pressure of the selected gas is set and the pellet sample is exposed to the laser beam for 10 s. The analysis of the mass spectrometer is set to begin with the start of the irradiation of the sample. A scheme of this setup is presented in Figure 16.

Different detectors can be used on the MID (Multiple Ion Detection) mode of the MS. Chemical species of atomic mass lower than 46 are followed using the usual Faraday detector. Intensities of species with *m/z* higher than 46 are followed using the Secondary Electron Multiplier (SEM) function of the MS. A secondary electron multiplier offers numerous advantages as it increases the sensitivity of the measurement method. Consequently, lower partial pressures can be detected.

### 3.3. Flash-Pyrolysis Setup (Py-GC-MS)

Propellant pyrolysis is also investigated using a flash pyrolysis apparatus (CDS, 5200 HP) linked to a gas chromatography (Varian, 3800) and coupled to a mass spectrometer (Varian, Saturn 4000 with ion trap, Santa Clara, CA, USA). The propellant sample is placed into a quartz tube and is heated by means of a platinum coil. The heating rate of the coil can go up to 20,000 °C·sec^−1^. The probe temperature is first set at 30 °C during 5 seconds before rising at 300, 500 or 700 °C with a rate of 8000 °C·sec^−1^. Maximum temperature is maintained during 15 s. During the heating phase, sample is under air. As the device is used in “trap-mode”, all the pyrolysis gases are collected in a triple-bed-trap (CDS, 90 mg-60:80-Tenax-TA™/Carboxen 1000/Carbosieve™ SIII, Oxford, PA, USA) before being flushed to the gas chromatography. The transfer line between the flash pyrolysis apparatus and the GC is heated at 280 °C to prevent any gases condensation. Helium is used as carrier gas with a constant flow rate of 30 mL·min^−1^. Injector (type 1177) temperature is set at 194 °C. The GC column is heated at 25 °C during 30 s and up to 35 °C at 0.5 °C·min^−1^. A VF-5ht (Varian, 30 m × 0.3 mm × 0.45 mm) column is linked to the mass spectrometer, and a VF-5ms (Varian, 30 m × 0.25 mm × 0.39 mm) column is connected to the flame ionization detector (FID).

Experiments are conducted as follows: a small piece of propellant pellet (average mass 0.8 mg) is placed into a quartz tube and inserted around the platinum coil. After the fast heating of the coil, the pyrolysis gases are collected in the flash pyrolysis trap. Heating rate is set at 8000 °C·s^−1^ to reproduce the heating rate of a 10 W laser at the surface of the pellet. Afterwards, the trap is heated at 250 °C to help desorption of all the absorbed species on the trap surface. Then, emitted gases are analyzed with the GC/MS.

### 3.4. Laser Ignition Setup

Laser ignition experiments are performed in a stainless-steel cylindrical reactor with an internal volume of 55 cm^3^. This reactor is equipped with a pressure sensor (Kistler 603B, Winterthur, Switzerland) and coupled to a laser diode through an optical system. The pressure sensor signals were recorded with a digital oscilloscope (Agilent Technologies, DPO2014B, Santa Clara, CA, USA). The laser diode and the optical system are the same as for the laser-induced pyrolysis experiments. A scheme of this setup is presented in Figure 17.

Two initial atmospheres are studied: nitrogen and argon, with initial pressures ranging between 25 and 70 bar. Laser temperature is fixed at 20 °C and 4 intensities are studied between 15 and 22 A, corresponding at this temperature to laser powers of 5.09, 6.54, 8.74 and 10.13 W.

Experiments are conducted as follows: sample propellant pellet is placed in a PMMA holder, and this holder is placed in the reactor and sealed with a sapphire window. Selected initial pressure of the selected gas is set, and sample is irradiated at the selected laser power during a selected time (corresponding to the energy). Pressure is recorded as a function of time if ignition occurs. For each experimental condition, experiments are performed in triplicate.

## 4. Conclusions

Pyrolysis, laser ignition and combustion of three low-vulnerability propellants made of RDX and NC were experimentally investigated in this paper.

Pyrolysis was found to be highly dependent on the surrounding gases and the initial pressure. Indeed, results have shown that increasing pressure enhances pyrolysis product emission. It was also noticed that argon enhances pyrolysis gases production, compared to nitrogen. Interestingly, it has been noticed that nitrogen could be an enhancer for the formation of hydrogen cyanide while argon is not. The Py-GC-MS experiments show the same trends as the laser induced pyrolysis. The probe temperature has a major influence on the results. Indeed, for various probe temperatures, the peak area extracted from FID signal are completely different.

Laser ignition experiments have shown that the 90-10 propellant is the one with the highest values of overpressures and propagation rates for each laser power studied. Regarding the ignition delays, the effect of nitrocellulose was obvious as the 85-15 composition gave the lowest ignition delays. Similarly to pyrolysis results, ignition experimental results have shown that argon enhances the combustion properties, leading to a higher propagation rate and overpressure than nitrogen.

For both pyrolysis and ignition experiments, an unexpected behavior was noticed for the 90-10 propellant, meaning that an optimum rate between RDX and NC could exist in order to improve combustion. The lack of kinetic data for a detailed decomposition scheme of NC is a major “brake” for theoretical predictions of thermal decomposition of such low-vulnerability propellants. Further investigations are therefore under progress concerning the kinetic modelling of the thermal decomposition of NC.

## Figures and Tables

**Figure 1 molecules-25-02276-f001:**
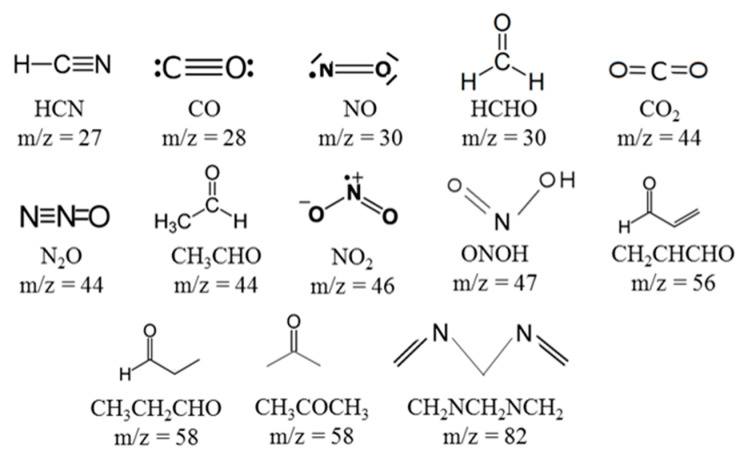
Pyrolysis species followed with MS.

**Figure 2 molecules-25-02276-f002:**
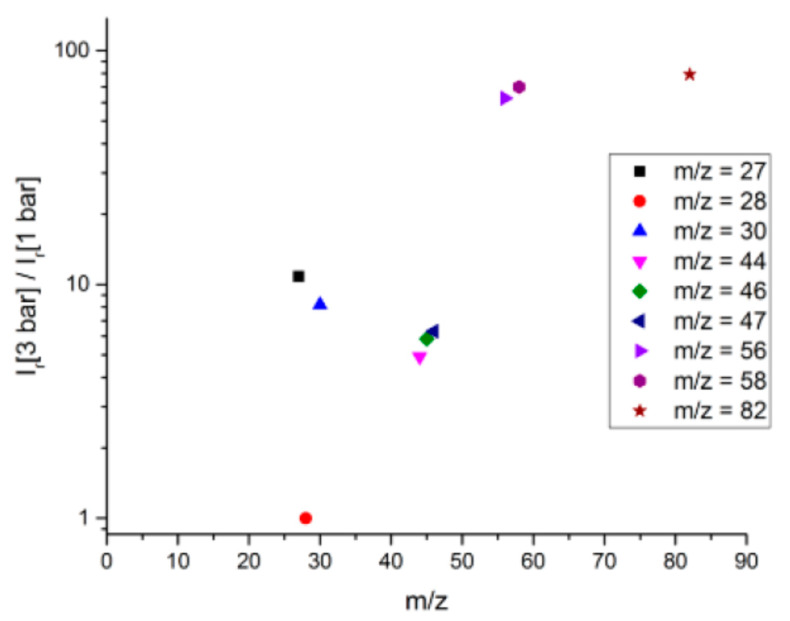
Quotient of the relative intensities at 3 bar and 1 bar of the emitted species during laser-induced pyrolysis of 90-10 propellant under nitrogen atmosphere.

**Figure 3 molecules-25-02276-f003:**
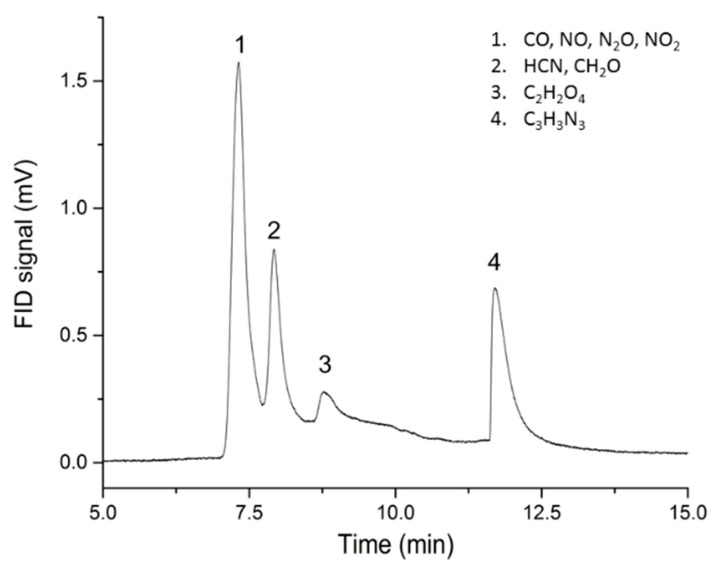
GC chromatograph for a probe temperature of 700 °C (85-15 propellant).

**Figure 4 molecules-25-02276-f004:**
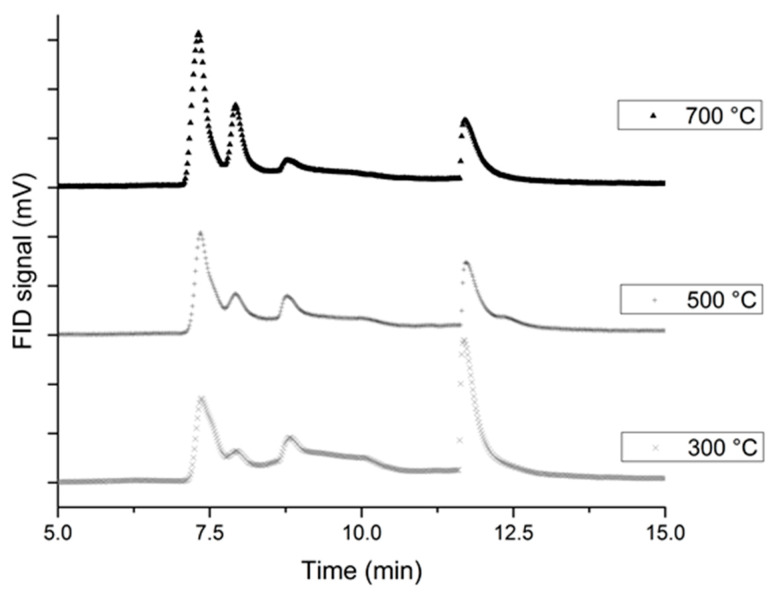
GC chromatographs regarding the probe temperature for the 85-15 propellant.

**Figure 5 molecules-25-02276-f005:**
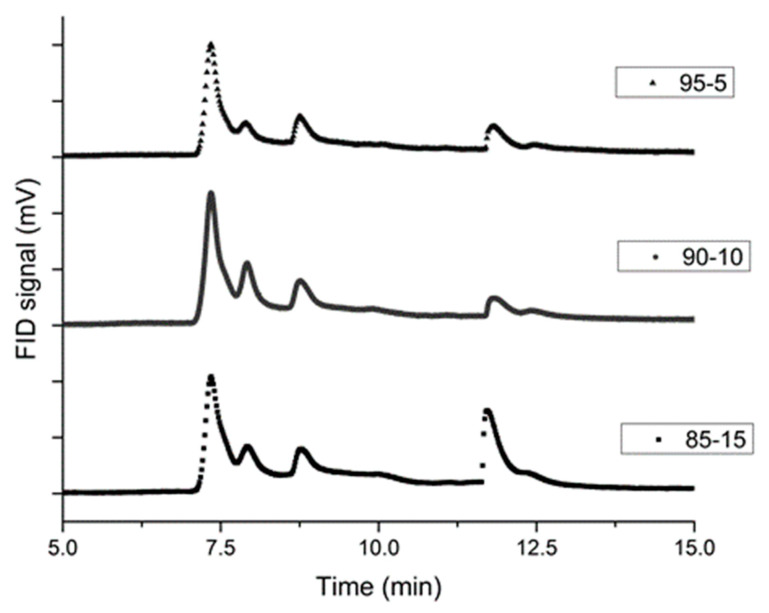
GC chromatographs of the 3 propellants for a probe temperature of 500 °C.

**Figure 6 molecules-25-02276-f006:**
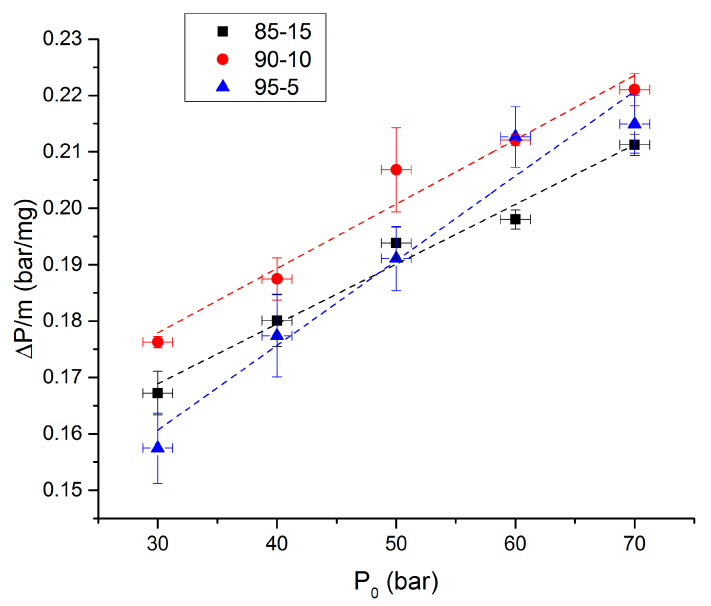
Overpressures as functions of initial pressures under argon.

**Figure 7 molecules-25-02276-f007:**
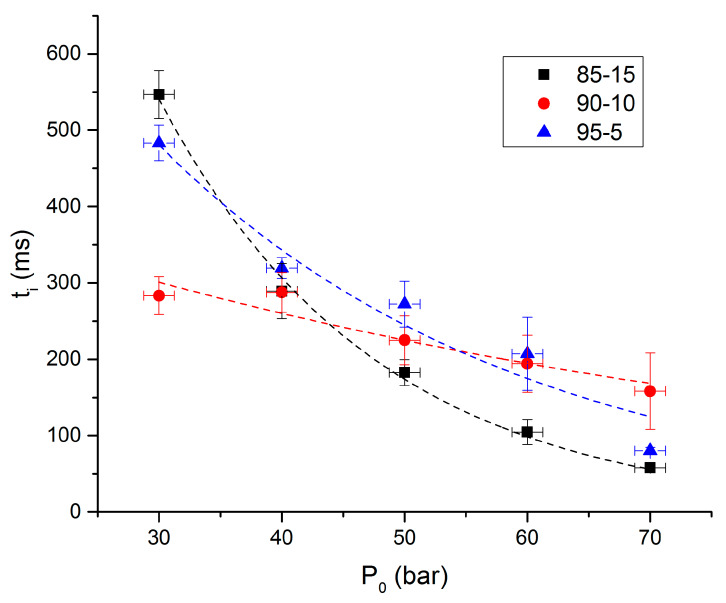
Ignition delays as functions of initial pressures under argon.

**Figure 8 molecules-25-02276-f008:**
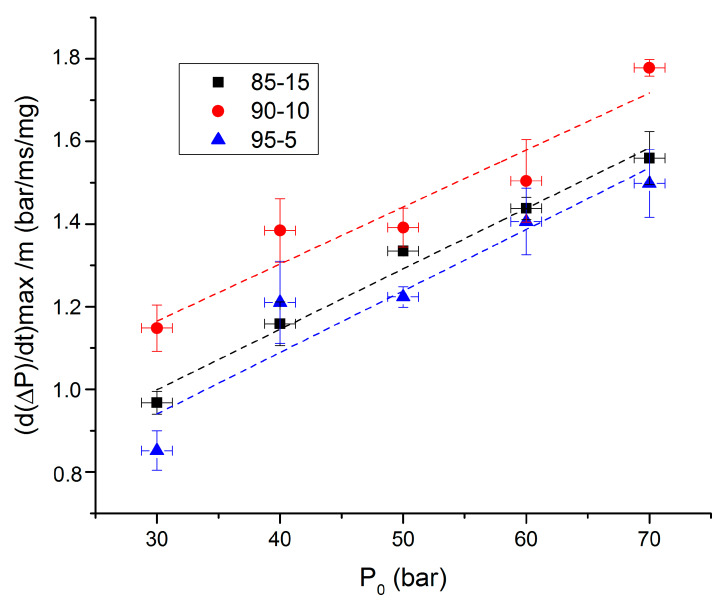
Propagation rates as functions of initial pressures under argon.

**Figure 9 molecules-25-02276-f009:**
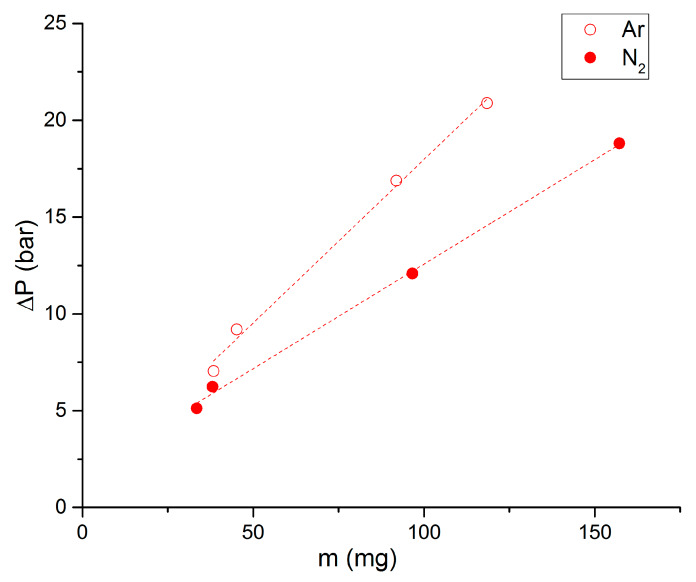
Overpressure as a function of initial mass, propellant 90-10, initial pressure of 50 bar.

**Figure 10 molecules-25-02276-f010:**
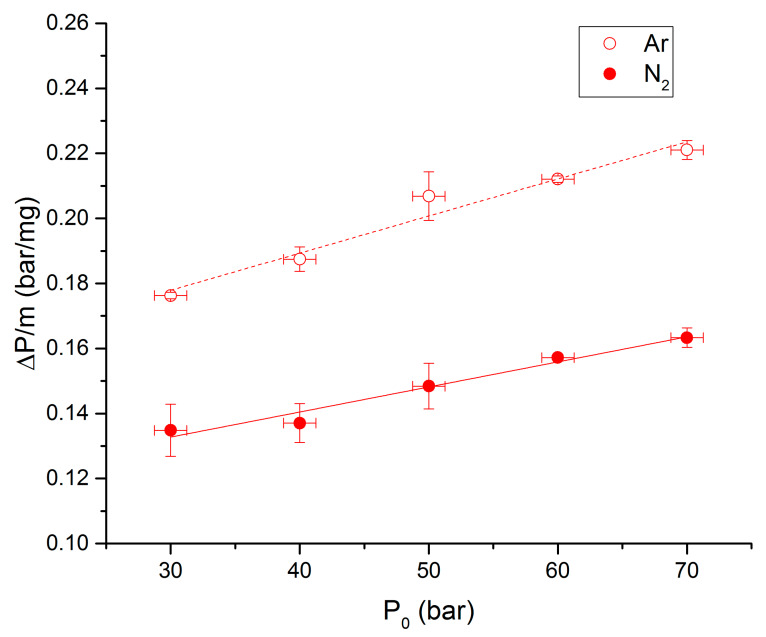
Overpressure vs. P0 for the 90-10 propellant under N_2_ and Ar.

**Figure 11 molecules-25-02276-f011:**
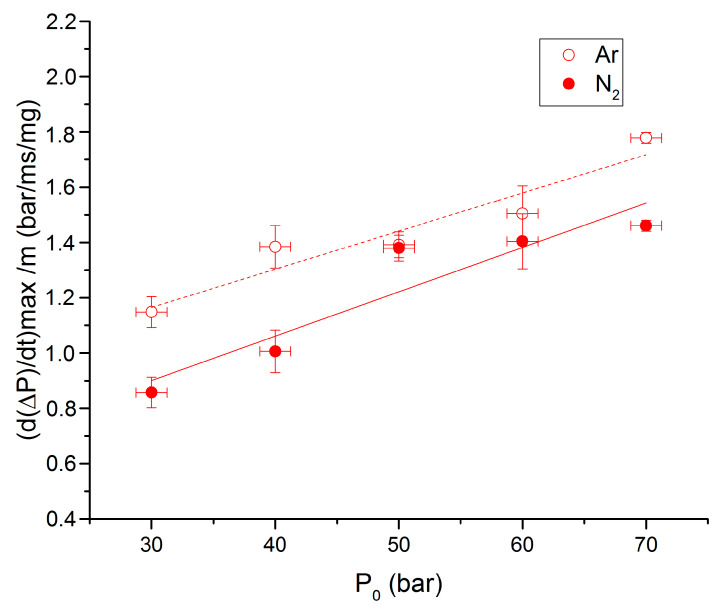
Propagation rate vs. P0 for the 90-10 propellant under N_2_ and Ar.

**Figure 12 molecules-25-02276-f012:**
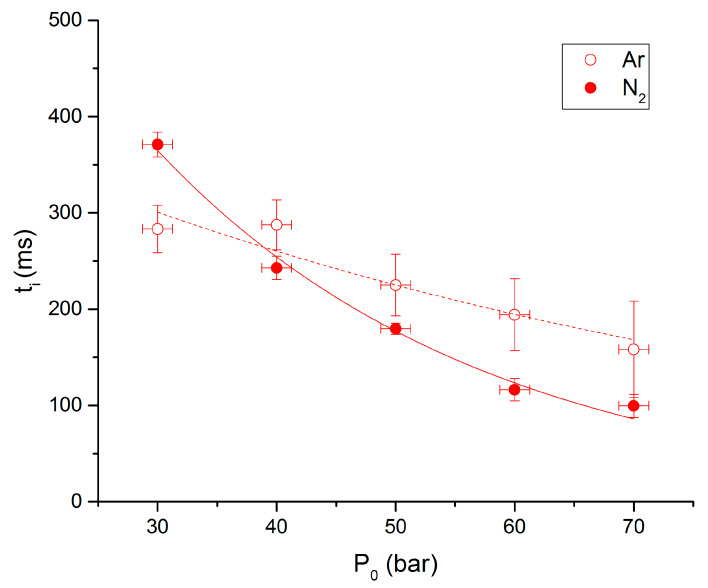
Ignition delay vs. P0 for the 90-10 propellant under N_2_ and Ar.

**Figure 13 molecules-25-02276-f013:**
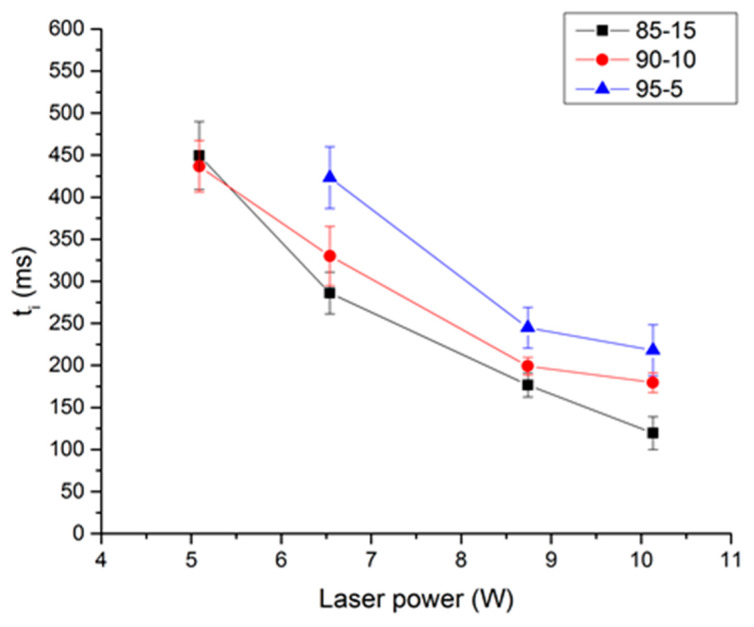
Ignition delay as a function of laser power for the three propellants, P0 = 50 bar of N_2_.

**Figure 14 molecules-25-02276-f014:**
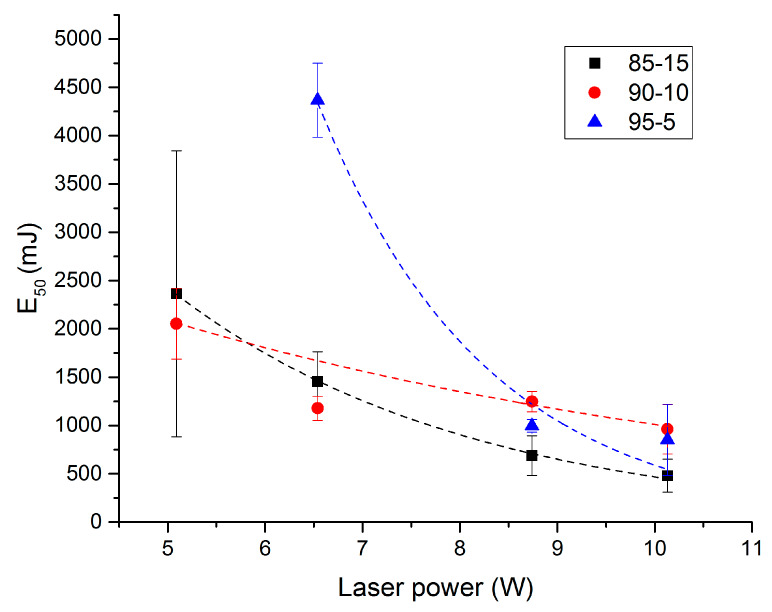
Ignition probability of 50% (E_50_) as a function of laser power for the three propellants, P0 = 50 bar of Ar.

**Figure 15 molecules-25-02276-f015:**
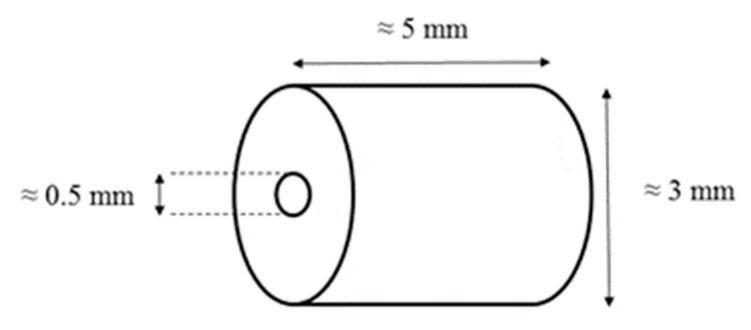
Scheme of the used propellant.

**Figure 16 molecules-25-02276-f016:**
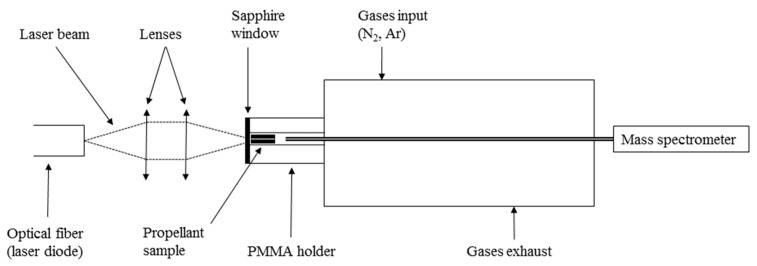
Scheme of the laser-induced pyrolysis setup.

**Figure 17 molecules-25-02276-f017:**
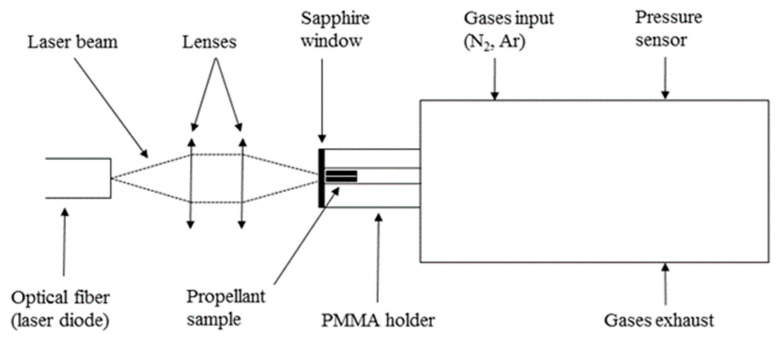
Schematic overview of the laser ignition setup.

**Table 1 molecules-25-02276-t001:** Pyrolysis species relative intensities regarding the gas nature.

		Argon Atmosphere (1 bar)			Nitrogen Atmosphere (1 bar)		
Detector	*m/z*	85-15	90-10	95-5	85-15	90-10	95-5
Faraday Cup	27	5.23 × 10^−3^	5.23 × 10^−3^	4.36 × 10^−3^	1.10 × 10^−2^	7.72 × 10^−3^	7.92 × 10^−3^
	28	1.01 × 10^−1^	1.46 × 10^−1^	3.06 × 10^−2^	1.00	1.00	1.00
	30	8.30 × 10^−1^	1.21 × 10^−1^	1.71 × 10^−2^	1.14 × 10^−2^	1.68 × 10^−2^	1.58 × 10^−3^
	40	1.00	1.00	1.00	*	*	*
	44	7.18 × 10^−2^	1.05 × 10^−1^	1.63 × 10^−2^	9.17 × 10^−3^	1.38 × 10^−2^	2.63 × 10^−3^
Electron Multiplier	46	4.65	6.80	9.21 × 10^−1^	6.59 × 10^−1^	1.11	2.53 × 10^−1^
	47	1.01 × 10^−1^	7.51 × 10^−2^	3.61 × 10^−2^	2.29 × 10^−2^	1.59 × 10^−2^	8.10 × 10^−3^
	56	1.56	2.13 × 10^−1^	2.47 × 10^−1^	2.21 × 10^−1^	8.05 × 10^−2^	1.55 × 10^−2^
	58	2.49	3.46 × 10^−1^	1.60	1.01 × 10^−1^	3.89 × 10^−2^	6.48 × 10^−3^
	82	5.27 × 10^−1^	2.00·× 10^−2^	5.10 × 10^−2^	9.82 × 10^−2^	2.28 × 10^−2^	2.16 × 10^−3^

* atomic mass unit not followed for these experiments.

**Table 2 molecules-25-02276-t002:** Coefficient values for the exponential regression t_i_ = a × exp(b × P_0_) for the 90-10 propellant.

	Ar	N_2_
**a** (ms)	4.64 × 10^2^	1.08 × 10^3^
**b** (bar^−1^)	−1.45 × 10^−2^	−3.61 × 10^−2^
**R^2^**	0.878	0.989

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
