# Peer review of "Experimental Study of Pyrolysis and Laser Ignition of Low-Vulnerability Propellants Based on RDX"

_molecules, 2020, doi:10.3390/molecules25102276_

Round 1

Reviewer 1 Report

The present manuscript deals with the kintics of pyrolisis of nitrocellulose-RDX formulations. To this end, the authors employed several modern experimental techniques. The reported results are interesting and sound, however, some improvements of the manuscript are neccessary before publication. The most important issues are:

  1. In the present form, it is more like a technical report. First, it would be more convinient for a reader to see the methodology section first, and then the results. The detected species (Fig. 16) are the part of Results, not methodology. In the Table 1, the physical methods, not the detector types (Faraday and SEM) should be mentioned. The absolute intensities do not have much sence without the apparatus calibration and even without the units. It is more reasonble to present the normalized intensities.
  2. It is better to discuss more physically related values, not "FID signal" and "overpressures". The formula (1) is actually not a formula.
  3. The properties of NC are tremedously sensitive to the nitrogen content and a percentage of water. The grades of NC used should be mentioned. The composition of Centralit 1 should also be explained for a wide readership of the journal.
  4. More discussion on the difference between the argon and nitrogen atmospheres. Is it possible to quantify the heat capacity differences? How to extrapolate these results to an air atmospere under normal conditions?
  5. Is it possible to describe the ignition delays observed with any type of kinetic equations? 
  6. Sometimes the wording is a bit weird, e.g, "thanks to". The lengthy abstract contains a too detaild description of the experimental conditions but do not list, e.g., any simple products detetcted.
  7. Making the bottom line: the manuscript shoild be modified to include more discussions on physics of the raw data reported.

Reviewer 2 Report

The article contains new material; it lies in the field of Journal scientific interests and can be published after a small correction.

The article “Experimental study of pyrolysis and laser ignition of low-vulnerability propellants based on RDX” by Jordan Ehrhardt, Léo Courty, Philippe Gillard and Barbara Baschung contains new material; it lies in the field of journal scientific interests and can be published after a small corrections. The reviewer has not found serious drawbacks in the article. However, there are several   questions should be answered by the authors:

  1. In the section 3.1 «Materials» of the article, the authors have written: Lines 241-245. “Propellants studied in this paper are laboratory propellants produced at the French-German Research Institute of Saint-Louis (ISL), their composition is perfectly controlled. They consist of tubular pellet made of nitrocellulose (NC), hexogen (RDX) and Centralit I as stabilizer”.

The reviewer thinks that the authors should give the nitrogen content in the NC. This information is necessary so that the reader can independently calculate the coefficient of excess of oxidizing agent in the formulation.

  1. The reviewer pay attention that Figure 6, Lines 165 - 166, requires a more detailed discussion of the presented results.

The authors have written, Lines 167-169: “It is clear in Figure 6 that ignition delay decreases when initial pressure increases. The influence of the NC rate is clearly visible. The formulation giving the highest ignition delay is the 95-5 and the lowest is obtained for the 85-15. The trend of these results was expected”.

The reviewer fully agrees with the thesis, however, the ignition delay time at an external pressure of 30 bar of argon has a relationship: the ignition delay time has a dependence: formulation 90-10 <95-5 <85-15; at an external pressure of 50 bar of argon, the ignition delay time has a relationship: formulation 85-15 <90-10 <95-5; and at a pressure of 70 bar argon, the dependence of the ignition delay time has change again: 85-15 <95-5 <90-10.

The reviewer insists that the authors should explain to readers the reason of the changes in the order of the ignition delay time of the RDX-NC formulations with increasing external argon pressure.

  1. The authors presented the dependence of 50% ignition formulations (E50) as a function of laser power for the three propellants (P0 = 50 bar of Ar) in Figure 13.

The authors have written: lines 230-231, “The effect of the laser power on the E50 is clearly visible in Figure 13. The higher the laser power, the lower E50 is”.  And then, Lines 235-238:”The 85-15 propellant formulation presents the lowest E50 for the highest powers studied. These results show the important effect of the NC in an energetic material composition. Here again, we can notice the unexpected behavior of the 90-10 propellant, which presents the lowest E50 for low powers”. Indeed, if laser beam has power 5 or 6.5 W, the formulation 90-10 has the lowest E50, however, when laser beam has a higher power, the dependence which has been proclaimed by the authors looks unexpected: for a laser beam power of ~ 8.7 W and ~ 10.2 W, the dependence E50 from the content of NC in the formulations has not been monotonic in nature 85-15 <95-5 <90-10.

This fact needs an explanation for readers.

Reviewer 3 Report

The paper investigated the pyrolysis, ignition and combustion characteristics of low-vulnurability propellants which was made of three different weight ratios of hexogen and nitrocellulose. In principle, I am happy with the paper. It’s good organized and well written. The discussion and conclusions are supported by the data.

I have four issues with the paper, one of which is grammar issue for the authors.

  • In Paragraph 1 of Section 1 “Introduction”, it would be better if the progress of LOVA propellant is also detailed.
  • The highest peak intensities based on laser-induced pyrolysis are listed into Table 1, the mass spectrum lines of m/z =28 are suggested to supply for clearer demonstration.
  • What is the heating rate of laser-induced pyrolysis, which should be supplied in Section 2.1 to compare with flash pyrolysis?
  • P1, L42, “One the other hand” should be revised as “On the other hand”; P3 L114, P12 L329, the word “when” should be replaced by “while”; P13, L315, it should be “placed”.
